# Diagnostic and Prognostic Role of Extracellular Vesicles in Pancreatic Cancer: Current Evidence and Future Perspectives

**DOI:** 10.3390/ijms24010885

**Published:** 2023-01-03

**Authors:** Alberto Nicoletti, Marcantonio Negri, Mattia Paratore, Federica Vitale, Maria Elena Ainora, Enrico Celestino Nista, Antonio Gasbarrini, Maria Assunta Zocco, Lorenzo Zileri Dal Verme

**Affiliations:** Department of Internal Medicine and Gastroenterology, Fondazione Policlinico Universitario “A. Gemelli” IRCCS, Università Cattolica del Sacro Cuore, 8, 00168 Rome, Italy

**Keywords:** extracellular vesicles, pancreatic cancer, biomarkers, proteomics, microRNA, early diagnosis, liquid biopsy

## Abstract

Pancreatic cancer is one of the most aggressive tumors, with a dismal prognosis due to poor detection rates at early stages, rapid progression, post-surgical complications, and limited effectiveness of conventional oncologic therapies. There are no consistently reliable biomarkers or imaging modalities to accurately diagnose, classify, and predict the biological behavior of this tumor. Therefore, it is imperative to develop new and improved strategies to detect pancreatic lesions in the early stages of cancerization with greater sensitivity and specificity. Extracellular vesicles, including exosome and microvesicles, are membrane-coated cellular products that are released in the outer environment. All cells produce extracellular vesicles; however, this process is enhanced by inflammation and tumorigenesis. Based on accumulating evidence, extracellular vesicles play a crucial role in pancreatic cancer progression and chemoresistance. Moreover, they may represent potential biomarkers and promising therapy targets. The aim of the present review is to review the current evidence on the role of extracellular vesicles in pancreatic cancer.

## 1. Introduction

Pancreatic cancer (PC) is one of the most significant public health problems and one of the major clinical and scientific challenges due to its rising incidence, the absence of accurate early diagnostic biomarkers, the lack of preventive screening, and the paucity of effective treatments [1]. With a 5-year survival rate of 5–10% and an average survival rate of 5–6 months following diagnosis, it is estimated that PC will become the third leading cause of cancer in Europe and the second one worldwide by 2030 [2,3,4]. This is the consequence of a late diagnosis in most cases diagnosed, when the disease is already at a locally advanced or metastatic stage, resulting in a small proportion of patients who can benefit from curative surgery.

There are currently no effective biomarkers for the diagnosis of pancreatic precancerous lesions, the prediction of malignant transformation of PC precursors, or the early identification of invasive PC [5].

Serum carbohydrate antigen 19-9 (CA 19-9) is the most studied PC biomarker. However, the limited sensitivity, specificity, and positive predictive value of CA 19-9 preclude its use as a screening biomarker for the early detection of PC [6,7]. 

Nonetheless, it is an effective prognostic tool for the assessment of tumor response to neoadjuvant therapy [8]. It becomes crucial, therefore, to develop biomarkers of pancreatic lesions that not only provide the early identification of in situ cancer but also the targeted treatment of early alterations.

In recent years, a growing body of research has focused on the identification of novel specific serum biomarkers for the early diagnosis of PC, commonly identified under the term “liquid biopsy” [9]. In this field, extracellular vesicles (EVs) are one of the most intriguing and promising research topics. EVs are lipid bilayer-coated globular organoids produced from cell membranes that contain molecules expressed by the originating cell, such as proteins, nucleic acids, and lipids. EVs are released by all cells into the extracellular environment where they mediate intercellular communication. In the natural history of cancer, EVs are involved in multiple pathways of carcinogenesis, metastasis, and chemoresistance. Hence, they can be found in every biological fluid, making them a suitable tool as a non-invasive biomarker. Hence, several methods for the isolation and identification of EVs were developed in order to exploit this potential. Moreover, the study of EV proteomics and genomics allowed for the identification of components that play a role in tumorigenesis and cancer progression, thus representing possible EV-labeling biomarkers. In the natural history of cancer, EVs are involved in multiple pathways of carcinogenesis, metastasis, and chemoresistance [10,11].

The aim of this review is to summarize the evidence regarding the potential role of EVs in the diagnosis, disease monitoring, and prognostic estimation of PC.

## 2. Extracellular Vesicles in Pancreatic Cancer: Role in Carcinogenesis, Cancer Progression, and Chemoresistance

EVs are heterogeneous subcellular structures composed of a phospholipid bilayer membrane with membrane proteins and glycoproteins containing various endocellular material, including bioactive molecules of the origin cell (microRNA, mRNA, transcription factors, cytokines, growth factors, lipids) [12,13,14]. According to size and biogenesis, EVs are divided into exosomes, microvesicles (MVs), and apoptotic bodies. Exosomes range in size from 30 to 150 nm. Their formation begins with the endocytosis of the plasma membrane, proceeds within the multivesicular endosome, and then they are discharged by exocytosis into the extracellular environment. The microvesicles range in size from 50 to 1000 nm and are produced via plasma membrane budding [15,16]. The structure and biogenesis of EVs have been thoroughly revised elsewhere [11] and are presented in Figure 1. Techniques for the isolation and characterization of EVs are presented in Table 1.

After release, EVs diffuse in the extracellular environment, where they play a remarkable role in cell-to-cell communication. Several membrane proteins act as a ligand for host cell receptors and activate specific metabolic pathways [34]. Otherwise, EVs can be internalized via endocytosis or can fuse with the recipient cells, releasing their content into the host cell [35,36]. All these mechanisms influence metabolism and genome expression, are involved in physiologic functions, and can enhance pathogenic mechanisms, such as inflammation and carcinogenesis [15].

Exosomes produced by the initial site of the cancer can stimulate proliferation, survival, immunosuppression, and the formation of the pre-metastatic niche in tissue-specific target cells via autocrine, paracrine, and distant mechanisms [34,37,38,39].

As for pancreatic carcinogenesis, the evidence suggests that PC-derived EVs can regulate pancreatic functions and angiogenesis, and promote tumor growth, progression, invasion, and metastasis through a multitude of pathways [40] (Figure 2).

It was demonstrated that cancer cells can promote the progression of other cancer cells through an EV-mediated communicating network involving the activation of Yes1-associated transcriptional regulators (YAP) via LDL receptor-related proteins [41]. Moreover, other components of the tumor microenvironment (TME) can release EVs that contribute to tumor growth and progression. Leca et al. demonstrated that EVs derived from cancer-associated fibroblasts (CAFs) undergoing non-physiologic culture conditions were enriched in ANXA6 and stimulated tumor growth and aggressiveness probably by activating the ANXA6/LRP1/TSP1 complex [42].

EVs may also have an important role in favoring the epithelial–mesenchymal transition (EMT) through mediators such as long non-coding RNAs (lncRNAs) [43,44]. Other proof of this effect comes from other experiments: the exposure of stromal cells to ANXA+ EVs induced a cell switching to a mesenchymal phenotype through the activation of formyl peptide receptors (FPRs) [45].

EVs are also involved in the acquisition of invasiveness and migration of PC cells. In a study by Pirlog et al., it was demonstrated that EVs containing SUMOylated heterogeneous nuclear ribonucleoprotein A1 (hnENPA1) promoted lymphangiogenesis and lymph node metastasis [46]. As for the effects on endothelium, in vitro studies demonstrated that PC cell-derived EVs induced tissue factor activity and increased thrombin generation in umbilical vein endothelial cells (HUVEC). This may contribute to a procoagulant status and cancer-associated thrombosis [47].

Zhang and colleagues demonstrated that PC cell-derived exosomes can increase the expression of PDGFB in receiving cells, promoting a recruitment of pancreatic stellate cells (PSCs) and the development of metastasis by transferring the exosomal protein Lin28B to metastatic cancer cells [48].

PC-derived EVs are also able to regulate the immune response [49]. Indeed, PC-derived EVs enriched in integrins are associated with a lower expression of NKG2D, CD107a, TNF-α, and INF-γ in natural killer (NK) cells, decreasing their anti-tumoral activity [50]. EVs were also associated with the switch of macrophages to M2 immunosuppressive phenotypes through microRNA(miRNA/miR)-155-5p and the regulation of the EHF/Akt/NF-kB axis [51]. Furthermore, PC-derived EVs were correlated with the increased activity of ERK1/2 MAP kinases in mast cells and adenosine signaling mediated by adenosine A3 receptors. Through this pathway, activated mast cells may contribute to tumor progression releasing inflammatory mediators [52].

EVs are also involved in the development of chemoresistance. Through EVs, drug-resistant neoplastic cells can transfer escape mechanisms to other cells. EVs can also bind circulating monoclonal antibodies, reducing their efficacy on target cells [53]. In PC, a study demonstrated a correlation between exosomes containing a high quantity of miRNA-155 and the acquisition of resistance to gemcitabine by previously sensible cells [54]. Furthermore, when cancer-associated fibroblasts (CAFs) are exposed to gemcitabine and nab-paclitaxel, they increase the production of extracellular EVs. Interestingly, cancer cell lines treated with these EVs exhibited an enhanced survival rate and gemcitabine resistance after treatment [55].

## 3. Proteomics of Extracellular Vesicles in Pancreatic Cancer

EVs contain a large variety of proteins, both on their surface and inside their lipid bilayer. Hence, the analysis of EV proteins and proteomic profiles allows the different EV subpopulations to be distinguished. Indeed, this is the thread that has accumulated the larger body of evidence. In particular, surface proteins own a remarkable significance since they can be used as antigens for the isolation using specific antibodies. In fact, the identification of surface proteins is one of the most validated methods to attribute EVs to the parent cells. In fact, both the surface and the content of the EV reflect the cell that generated it. However, ideal tumor-specific biomarkers have not already been identified.

Melo et al. identified Glypican-1 (GPC1) as a potential tumor-specific protein from PC cell culture by a proteomic analysis of tumor-derived exosomes. GPC1 is a membrane-anchored protein only detected in cancer cells. They validated this observation in the serum samples of 190 patients with PDAC, who showed significantly higher levels of GPC1+ exosomes compared with patients with a benign pancreatic disease and healthy controls (*p* < 0.0001). Moreover, 100% of GPC+ exosomes from PDAC patients contained mutant *KRAS*, confirming they originated from neoplastic cells. Levels of serum GPC1+ exosomes distinguished patients with PDAC, either in early or late stage, from healthy subjects and patients with a benign pancreas disease, exhibiting absolute sensitivity (100%—95% CI: 98.1–100%) and specificity (100%—95% CI: 97.1–100%), with a positive predictive value of 100% (95% CI: 98.1–100%) and a negative predictive value of 100% (95% CI: 86.8–100%). Standard biomarkers, such as CA 19-9, were outperformed (AUC of 0.739, 95% CI: 70.2–82.6%, *p* < 0.001), being significantly elevated in patients with a benign pancreas disease (*p* < 0.0001). Levels of GPC1+ exosomes also correlated with tumor burden in patients with PDAC: patients with distant metastatic disease had significantly higher levels (average of 58.5%) when compared to patients with metastatic disease restricted to lymph nodes (average of 50.5%) or no metastases (average of 39.9%). In addition, GPC1+ exosomes levels before and after surgery stages were investigated in 29 PDAC patients: 28 of them showed decreased GPC1 levels after a surgical resection (*p* < 0.0001), while CA 19-9 levels decreased only in 19 patients (*p* = 0.003). Moreover, patients who presented a higher decrease in levels of GPC1+ exosomes showed increased survival compared with patients with a lower or no decrease in GPC+ exosomes (27.7 vs. 15.5 months). Conversely, serum CA 19-9 levels were not significantly associated to survival. Finally, in a Cox regression model which included the decrease in GPC1+ exosomes levels, median age, American Joint Committee on Cancer (AJCC) stage, tumor grade, and CA 19-9 levels, only GPC1+ exosomes level was an independent prognostic and predictive marker for disease-specific survival (hazard ratio (HR): 5.353, CI: 1.651–17.358, *p* = 0.005) [56].

The ability to distinguish PC from benign lesions using GPC1+ EVs was recently argued. Indeed, Frampton and colleagues confirmed the correlation between GPC1+ EVs levels and tumor burden: patients with higher GPC1+ EVs levels had significantly larger PDAC (>4 cm; *p* = 0.012) and in matched pre- and post-operative plasma samples there was a significant reduction of GPC1 levels after surgical resection for PDAC (97 ± 54 vs. 77.8 ± 32.4 pg/mL; *p* = 0.0428). However, there was no significant difference in plasma GPC1+ exosomes between both patients with PDAC and benign pancreatic diseases (chronic pancreatitis, IPMN, and serous cystadenoma). Similarly, GPC1 levels in PDAC and normal pancreatic tissues showed no differences. The sensitivity and specificity of plasma GPC1+ exosomes in the prediction of PDAC were 74% and 44%, respectively, with an area under the curve (AUC) of 0.59 [57]. 

Another recent study has reached similar conclusions: alone or in association with glycoprotein 2 (GP2), GPC1+ EVs were not adequately effective in distinguishing malignant from benign pancreatic lesions. The sensitivity and specificity of GPC1+ EVs for pancreatic adenocarcinoma were 26.67% and 87.50%, respectively. When combined with positivity for GP2, EV sensitivity and specificity were 23.33% and 90%, respectively [58].

Buscail et al. combined the diagnostic accuracy of circulating tumor cells (CTC—DAPI+, CK+, EpCAM+, and CD45−) or GPC1+ exosomes in peripheral and portal blood, reaching 45% (5/11) of accuracy in the identification of PDAC in portal blood and 10% (2/22) in peripheral blood. Importantly, the concomitant detection of CTC and GPC1+ exosomes displayed a sensitivity of 100% and a specificity of 80%, with a negative predictive value of 100%. High levels of GPC1+ exosomes and/or CTC presence were significantly correlated with progression-free survival and overall survival when CTC clusters were found [59].

The zinc transporter ZIP4 was found to be upregulated on the membrane of exosomes derived from a PC cell lines medium. Comparing the exosomal protein profiles of two cell lines, PC-1.0 (highly malignant) and PC-1 (moderately malignant), Jin and colleagues observed that PC-1.0-derived exosomes were more enriched in proteins that play key roles in cancer progression. In particular, ZIP4 was the most upregulated protein in PC-1.0-derived exosomes. The authors also compared the diagnostic value of serum ZIP4+ exosomes between patients with PC (n = 24), benign pancreatic diseases (n = 32, AUC = 0.89), biliary diseases (n = 32, AUC = 0.8112), and healthy controls (n = 46, AUC = 0.8931). In the experiment, malignant cells were able to transfer their oncogenic potential to benign ones through exosomes, accelerating the progression of the disease. The blockade of this trafficking mediated by EVs might also reduce tumor progression, consequently increasing survival [60].

Proteomic analysis of the exosomal “surfaceome” revealed multiple possible PDAC-specific biomarker candidates, such as CLDN4, EPCAM, CD151, LGALS3BP, HIST2H2BE, and HIST2H2BF. Exosomes were isolated from the blood samples of 103 PDAC patients. In the whole population of exosomes, *KRAS* mutations were detected in 44.1% of patients undergoing active therapy, whereas 73% of exosomes expressing selected biomarkers showed *KRAS* mutations [61].

Another study demonstrated that the presence of the serum Annexin A6+ (ANXA6) EVs is associated with PDAC, representing a potential biomarker. Interestingly, increased PDAC aggressiveness was associated with the tumor cell-mediated uptake of CAF-derived ANXA6+ EVs carrying the ANXA6/LRP1/TSP1 complex, whereas the depletion of ANXA6 in CAFs impaired the complex formation and subsequently the occurrence of metastasis [42].

EVs may play a crucial role in the progression of cancer and development of metastasis. Several proteins have been implicated in the formation of the metastatic niche [62]. CD151 and Tspan8 recruit and activate integrins, modulating the extracellular matrix, and facilitating motility and invasiveness. High levels of EVs expressing these mediators correlate with advanced disease: they may be essential for exosome-initiated target cells—cells that are supposed to account for tumor progression and metastatic settlement—and non-cancer initiating cells activation and reprogramming [63].

Furthermore, in the liver, which is the main target organ for PDAC metastasis, Kupffer cells (KCs) modify the microenvironment in response to the stimulation of regulatory cytokines and other factors. Indeed, a high concentration of macrophage migration inhibitory factor (MIF) promotes angiogenesis and it is also implicated in the loss of polarity and cell adhesion [64]. In particular, exosomal MIF induces the release of transforming growth factor β (TGFβ) by KCs, stimulating fibronectin (FN) production by hepatic stellate cells (hSCs), and contributing to the formation of the pre-metastatic niche. The upregulation of macrophage migration inhibitory factor (MIF) may be an early event of cancer progression: in fact, markedly higher levels of this cytokine were detected from plasma exosomes of stage I PDAC patients who later developed liver metastasis compared with patients whose pancreatic tumor did not progress [38].

More recently, since the analysis of EVs requires multiple laboratory steps, assays to test groups of EVs have been developed in order to facilitate the use of this promising tool in clinical practice. The main aim is to find out new strategies of analysis that require a minimal amount of biologic fluids, short time of processing, and are usually amenable to automation, reducing all those elements that limit their routine clinical application.

A rapid, ultrasensitive, and inexpensive nanoplasmon-enhanced scattering (nPES) assay that directly quantifies tumor-derived EVs from as little as 1 μL of plasma was used by Liang et al. to identify a possible PC EV biomarker, ephrin type-A receptor 2 (EphA2). This technique showed a good accuracy in the differentiation of PC patients from patients with pancreatitis (AUC 0.96) and healthy subjects (AUC 0.93). The levels of EphA2+ EVs correlated with staging and tumor progression, with a discriminatory sensitivity only modestly diminished for early stages of PDAC vs non-cancerous diseases (91%) or pancreatitis (86%) comparisons. In addition, plasma EphA2+ EVs were useful tools to predict PC responses to neoadjuvant therapy, being significantly decreased in patients with good/partial therapy responses but not in patients with poor responses. Their performance was better than CA 19-9, whose levels did not significantly differ between the groups in terms of the response to treatments. Thus, this biomarker may be considered a prognostic indicator to monitor PC patients’ responses to therapy [65]. 

There is some evidence that quantitative and qualitative abnormalities in the glycosylation of proteins play a role in carcinogenesis and tumor progression [66]. CA 19-9, the most validated biomarker for PDAC, is a carbohydrate antigen. Therefore, glycomic profiling is gaining interest, particularly through the analysis of interactions between multiple lectins and carbohydrates, by using a method called “lectin microarray system”. The detection of disease-specific glycosylation changes may allow for the identification of novel biomarkers for early diagnosis of PDAC. Particularly, several glyco-biomarkers, such as O-linked glycosylation on Mucin-1 (MUC1), have been successfully identified using the lectin microarray system in various diseases [67]. For example, the detection of specific alterations of MUC1 glycosylation allowed patients with cholangiocarcinoma, hepatolithiasis, and normal controls (*p* < 0.0001) to be distinguished, verifying the cholangiocarcinoma-related glycosylation changes by the lectin-antibody sandwich ELISA. 

In addition, Yokose and colleagues explored the glycomic profile of EVs. The authors evaluated the expression of O-glycan-binding lectins Agaricus bisporus agglutinin (ABA) and Amaranthus caudatus agglutinin (ACA), which were elevated specifically in PDAC sera, including those of patients at early stages. In particular, O-glycan-binding lectin+ EVs were significantly higher in the preoperative blood samples of PDAC patients than healthy controls (*p* < 0.001 and *p* < 0.001, respectively). The levels of labeled EVs were significantly reduced in the post pancreatectomy sera, at comparable levels to those of healthy controls (*p* < 0.001 and *p* < 0.001, respectively) [68].

## 4. EV-Derived miRNAs as Potential Biomarkers for Pancreatic Cancer

In a complementary fashion to proteomic analysis, EV-derived nucleic acids have been correlated with clinicopathologic features and prognosis, showing an interesting potential as biomarkers for the early diagnosis and prognosis of PC. In particular, a remarkable interest in the study of miRNAs grew in the last few years. miRNA are short non-coding RNA sequences (usually 20–23 nucleotides) that are produced from a precursor transcript by consecutive cleavage. They can target mRNA or other miRNA, inhibiting their expression. miRNA targeting of negative regulators of oncogenic pathways may result in an oncogenic effect. Similarly, the reduction of miRNA-targeting oncogenes may enhance carcinogenesis. For these reasons, several studies have investigated their potential role as biomarkers and therapeutic targets in cancer [69]. The encapsulation of miRNA into EV prevents their degradation, prolonging their half-life. Hence, EV-derived miRNA seem to be more reliable biomarkers compared with cell-free blood ones [70].

miR-192-5p may act as a tumor suppressor on epithelial-to-mesenchymal transition through the downregulation of the transcription factor, ZEB2. Flammang et al. demonstrated that it predicts PDAC with a diagnostic accuracy comparable to CA 19-9. In the same study, the expression of miR-192-5p was able to distinguish PDAC patients from healthy controls, both in tissue samples (AUC = 0.86, *p* < 0.0001) and exosomes (AUC = 0.83, *p* = 0.0004), as well as healthy controls from patients with chronic pancreatitis (tissue samples AUC of 0.80; *p* = 0.0021 – exosomes AUC of 0.80; *p* = 0.0164). However, unlike CA 19-9, it was not able to distinguish between patients with PDAC and chronic pancreatitis, both in tissue-derived and exosomal samples [71].

Reese and colleagues investigated the use of a panel of 11 miRNAs, consisting of miR-21, -34a, -99a, -100, -125b, -148a, -155, -200a, -200b, -200c, and -1246, to predict their association with proliferation, epithelial-to-mesenchymal transition, and chemoresistance in tissue and blood serum specimens of PDAC patients. Interestingly, they found an overexpression of miR-200b and miR-200c in the serum exosomes of PDAC patients compared with healthy controls (*p* < 0.001; *p* = 0.024) and patients with chronic pancreatitis (*p* = 0.005; *p* = 0.19). The expression of miR-200b and miR200c from total and EpCAM+ serum exosomes combined with the elevation of serum CA 19-9 resulted in a diagnostic accuracy of 97% (*p* < 0.0001) in the prediction of PDAC. In addition, higher levels of miR-200c in total serum exosomes (*p* = 0.038) were associated with shorter overall survival. In this study, EpCAM+ exosome-derived miR-200b was an independent prognostic factor for PDAC (*p* = 0.044) [72].

In another study, miR-200b and miR-200a were significantly elevated in the sera of PC and chronic pancreatitis patients compared with healthy controls (*p* < 0.0001), with a mutual correlation between miR-200, SIP1, and E-cadherin expression. In particular, miR-200b distinguished patients with PDAC from healthy controls with a sensitivity and specificity of 71.1% and 96.9%, respectively [73].

It has also been demonstrated that exosomal miRNA(exmiR) -21 may differentiate patients with early-stage PDAC from healthy controls. A tethered cationic lipoplex nanoparticle biochip was used to directly analyze exmiRs within a single exosome. Moreover, when combined with exmiR-10b, its diagnostic value improved (AUC = 0.791, *p* < 0.0001) in the differentiation of patients with early-stage PDAC from controls and advanced-stage PDAC (*p* < 0.05, early stage vs. healthy; *p* < 0.001, early stage vs. advanced stage) [74].

A concomitant evaluation of the exosomal surface antigen panel PaCIC (CD44v6, Tspan8, EpCAM, MET, and CD104) and serum exosome miRNA markers (miR-1246, miR-4644, miR-3976, and miR-4306) significantly improved sensitivity (1.00, CI: 0.95–1) with a specificity of 0.80 (CI: 0.67–0.90), when PDAC patients were compared with all others groups, and of 0.93 (CI: 0.81–0.98) when non-pancreatic malignancies were excluded [75].

Que et al. detected higher levels of EV-derived miR-17-5p in serum samples of patients with PDAC compared with chronic pancreatitis and healthy controls. A significant correlation with metastasis and advanced stage PDAC was also observed. However, they were not significantly correlated with PDAC different stages, differentiation, or tumor stage (*p* = 0.339, 0.385 and 0.668, respectively) [76].

## 5. EVs and EV-Derived miRNAs in Pancreatic Precancerous Lesions

Pancreatic precancerous precursors are dysplastic lesions with a potential risk of developing invasive carcinoma. These lesions are pancreatic intraepithelial neoplasia (PanIN), mucinous cystic neoplasm (MCN), and intraductal pancreatic mucinous neoplasm (IPMN). While PanIN is a microscopic condition, MCN and IPMN are frequently discovered incidentally with imaging techniques that are performed for other indications and patients are often asymptomatic at clinical presentation. It has been hypothesized that mucinous pancreatic precancerous lesions are responsible for up to 15% of PDAC [77]. Despite the recommended surveillance strategies, there is an urgent need for biomarkers that are able to predict the progression of precancerous lesions to malignancy early. These tools would sharpen the appropriateness of surgical treatments, preventing under- and over-treatments [78,79].

Melo et al. observed that GPC1+ EVs were also significantly higher in patients with histologically proven IPMN (n = 5) undergoing surgery because of clinical symptoms or evidence of a macroscopic mass in comparison with healthy donors and a benign pancreatic disease control group (18 patients with pancreatitis and 8 with serous cystic adenomas). At histopathology, two patients had IPMN with associated carcinoma in situ, one had an IPMN with early adenocarcinoma (pT1), one had IPMN with intermediate dysplasia, and one had IPMN with low-grade dysplasia. The levels of GPC1 in healthy donors and the benign pancreatic disease control group were similar [56]. Using immunohistochemistry, GPC-1 was expressed diffusely in PanIN (87.0%, 20/23) in over a half of IPMNs (58.1%, 79/136, particularly gastric-type, oncocytic-type, and pancreatobiliary-type IPMNs) and in a minority of MCNs (25%) [80]. However, a control population with pancreatic precancerous lesions was not tested in other studies evaluating GPC1+ EVs in pancreatic diseases [58].

In a recent study by Yang et al., a wide panel of 22 EV biomarkers (MUC1, MUC2, MUC4, MUC5AC, MUC6, Das-1, STMN1, TSP1, TSP2, EGFR, EpCAM, GPC1, WNT-2, EphA2, S100A4, PSCA, MUC13, ZEB1, PLEC1, HOOK1, PTPN6, and FBN1) was used to detect the development of PC in a plasma-based discovery cohort (n = 86) including healthy, age-matched benign controls and patients harboring low-grade (LG), high-grade (HG), and invasive/high-grade (inv/HG) IPMN. Several of these 22 biomarkers were elevated in patients harboring both LG and HG lesions; only secreted mucin MUC5AC was significantly higher in HG lesions. Among the 11 patients with invasive IPMN, 9 had high expression of MUC5AC in plasma EV, while among the 11 patients with high grade dysplasia alone, only 1 had high MUC5AC expression. MUC5AC exhibited a sensitivity of 100%, a specificity of 82%, and a diagnostic accuracy of 96% in differentiating inv/HG IPMN from LG-IPMN in the training cohort. In the validation cohort (N = 44), MUC5AC expression was observed in all three cases harboring inv/HG IPMN. In the combined cohort, MUC5AC exhibited 97–100% specificity with 33–50% sensitivity and an AUC of 0.65–0.73 in identifying invasive IPMNs. The use of MUC5AC+ EV diagnosed inv/HG IPMN in 36% of cases that would have been otherwise missed by imaging alone, giving MUC5AC great potential as a biomarker for improved clinical decision-making in patients at high risk of PC lesions [81]. While MUC5AC expression has been observed in all histological subtypes of IPMN, representing an essential criterion for their classification, it is intriguing that its presence has only been observed in circulating EVs of patients with INV/HG IPMN. It is unknown whether it is a consequence of increased tissue expression of MUC5AC in INV/HG lesion or a result of increased EV biogenesis, secretion, or selective packaging of MUC5AC in EVs [82].

Not only surface proteins, but also EV miRNAs have been used as potential biomarkers for IPMN malignant transformation. In a study by Yuki et al., miRNAs were extracted from serum EVs of 38 patients with IPMN (11 with invasive IPMN and 27 with benign IPMN) and 21 non-tumor controls. The results of the microarray analysis revealed that the expression of EV miR-22-3p, EVmiR-4539, and EVmiR-6132 were higher in patients with benign IPMN and invasive IPMN serum samples compared with the control group. Receiver operating characteristic (ROC) analysis showed that five markers may discriminate patients with IPMN and from control patients, with areas under the curve (AUCs) of 0.72 for EV miR-4539 (95% confidence interval [CI]: 0.59–0.85), 0.55 for carcinoembryonic antigen (CEA) (95% CI, 0.39–0.71), 0.55 for CA19–9 (95% CI, 0.41–0.71), 0.59 for EV-miR-22 (95% CI, 0.42–0.76), and 0.64 for EV miR-6132 (95% CI, 0.47–0.81). EV miR-4539 had the highest diagnostic yield compared with other markers at the cutoff value of 3.2 copies/μl, and the sensitivity and specificity were 60.5% and 95.2%, respectively, and exhibited a statistically significant difference (*p* = 0.004) in distinguishing IPMN from controls. 

ROC curve analysis revealed that the markers may distinguish patients with invasive IPMN from those with benign IPMN, with an AUC of 0.77 for EVmiR-6132 (95% CI, 0.610.93), 0.74 for CA 19-9 (95% CI, 0.520.97), 0.61 for EVmiR22 (95% CI, 0.410.81), and 0.58 for EVmiR-4539 (95% CI, 0.34–0.77). At the threshold value of 1.4 copies/l, EVmiR-6132 had the highest diagnostic yield compared to other markers, with a sensitivity and specificity of 88.3% and 65.4%, respectively. miR-4539 is also a promising biomarker for the identification of IPMN in the general population, as evidenced by an AUC of 0.72 derived from an examination of ROC curves. Additionally, ROC analysis showed that EV miR-6132 was able to discriminate individuals with invasive IPMN from those with benign IPMN with a diagnostic accuracy equivalent to CA 19-9 (AUC 0.77 vs. 0.74), with the added benefit that biliary obstruction did not affect expression levels [83].

Another study by Sonohara et al. analyzing sera from 63 patients (44 with PC, 15 with IPMN and 4 with mucinous cystic neoplasm) revealed that exosome-derived miR-196b and miR-204 are able to predict the prognosis of PC. miR-196b-3p and miR-204-3p in serum exosomes were differentially expressed among intraductal papillary mucinous neoplasms, mucinous cystic neoplasms, and PC. 

In particular, miR-196b-3p staining in PC tissues showed significantly higher expression compared with normal pancreatic tissues (*p* = 0.009), while miR-204-3p did not significantly differ (*p* = 0.85).

Serum exosomal levels of miR-196b-3p in PC patients were higher than IPMN patients’ (*p* = 0.14). No significant difference was found between mucinous cystic neoplasm and PC patients (*p* = 0.59). 

The expression of miR-204-3p was significantly higher in serum exosomes from patients with mucinous cystic neoplasm than patients with IPMN (*p* = 0.02, *p* = 0.007), while there was a marginally significant difference between mucinous cystic neoplasm and PC patients (*p* = 0.24). Therefore, miR-204 may be downregulated during the natural history of cancer progression, resulting relatively higher in early cancer stages and other low-malignant pancreatic tumors [84].

Goto et al. analyzed exmiRs in a population of 32 patients with PC, 29 patients with IPMN and 22 patients in the control group without malignant lesions. Among a total of 347-detected exmiR, the expression of exmiR-191, exmiR-21 and exmiR-451a was significantly increased in both the IPMN and PC groups than controls. The diagnostic performance of the three exmiRs was compared with those of CEA and CA 19-9. The levels of CA 19-9 were significantly higher in the PC group than the control or IPMN groups. At ROC analysis, the AUC and diagnostic accuracy of exmiRs were superior to those of CA 19-9 and CEA. In addition, ROC analysis between the control and earlier stages of PC including patients with stage I and IIa showed that the accuracy of the exmiRs was preferable compared with CEA. There was no significant difference in comparison with CA 19-9; however, the positive detection rate was approximately 10% higher. On the other hand, CA 19-9 was the best parameter for the diagnosis of advanced-stage PC (stage ≥IIb) (AUC 0.893, accuracy 90%). The exmiRs showed a good AUC value and accuracy (exmiR-21, AUC 0.862, accuracy 83.7%). Taken together, the results suggested that exmiR-191, exmiR-21, and exmiR-451a were good diagnostic markers for IPMN and early-stage PC, but that CA19-9 was still superior for the diagnosis of advanced cancer [85].

## 6. Future Perspectives

Despite progression being made during the last few years, further standardization and validation of EVs isolation techniques is still needed. Indeed, these methods are still significantly operator- and technology-dependent. Combining different techniques may help to limit the disadvantages of each one [86]. However, costs still represent a significant limitation. Moreover, the elucidation of biogenesis may allow the identification of more homogeneous subpopulations of EVs to be sharpened [87]. These results would significantly strengthen the quality of the observations, allowing more thorough mechanistic conclusions to be drawn.

The ability to Isolate and analyze several antigens is crucial to identify sensitive and specific biomarkers. The advent of next-generation biotechnological techniques, such as single-cell and single-EV analysis, high-throughput, and shotgun proteomics, may increase the ability to screen potential candidate biomarkers [88]. Moreover, the development of multiple array tests to contemporarily analyze several antigens may also enhance the diagnostic accuracy [89,90]. At the same time, the study of tumor EVs is useful for the identification of tumor-specific biochemical profiles and customize the therapy based on the biomolecular characteristics of the disease [91].

The definition of the diagnostic accuracy of EVs in the diagnosis of PC requires larger populations of patients and controls. Moreover, the majority of the studies lack external validation cohorts to confirm the results of the pilot studies. Considering that study populations are thoroughly selected, the application of EVs in the study of real life populations in clinical trials would allow the understanding of the real effectiveness of this promising tool. 

As for genomic analysis, deciphering the actions of miRNAs and lncRNA on oncogenes and tumor suppressor genes may allow the qualitative leap from simple association to causality to be made [69]. In this context, there is a growing interest in the study of extrachromosomal circular DNA (eccDNA). eccDNA has a complex biogenesis that is significantly enhanced in cancer cells [92,93,94]. It can transfer complete oncogenes from one cell to another, favoring cancer progression. Evidence in PC is limited to experimental cell lines [95]; however, the study of eccDNA may reveal important insights into PC carcinogenesis. At present, no study has analyzed EV-derived eccDNA, although EVs may represent a perfect cargo for the transfer of eccDNA to recipient cells. In addition, genetic testing for the oncogenes of EVs content in patients with precancerous lesions may help to stratify the risk of developing PC [77].

Finally, the isolation and identification of EVs still require highly qualified operators, a long period of time, and advanced machines. So far, the analysis of EVs has not left the experimental setting. Hence, the development of industrial kits would help the diffusion of these techniques and possibly the reduction of costs. Indeed, it is important to favor the application in clinical practice by easy-to-use tools [96,97]. Most of the EV isolation techniques involve several laboratory steps that cannot be patented. This limits the interest in investments by industrial companies. Nevertheless, some patents for innovative laboratory processes and machine-learning approaches were recently filed [98]. Patents could also be filed for devices and laboratory tools, whose development would certainly help the translational transition of EV application into clinical practice.

## 7. Conclusions

PC is a major health problem with an unmet and urgent need for the identification of biomarkers for the early diagnosis of the disease, the screening of populations at higher risk and the prognostic evaluation of patients’ response to treatments. Pathogenic mechanisms of carcinogenesis and tumor progression of PC are still incompletely understood. However, a growing body of evidence has raised the attention on the role of EVs in the pathogenesis of PC. Owing to their direct origin from neoplastic cells, the ability to diffuse in biologic fluids, and protection of the cargo, EVs are a candidate for promising biomarkers for PC. Unraveling the conundrum of the complex interaction between neoplastic, immune, and stromal cells mediated by EVs may finally allow for identifying specific PC-derived subpopulations of EVs for the early diagnosis of PC and the prognostic stratification of patients.

## Figures and Tables

**Figure 1 ijms-24-00885-f001:**
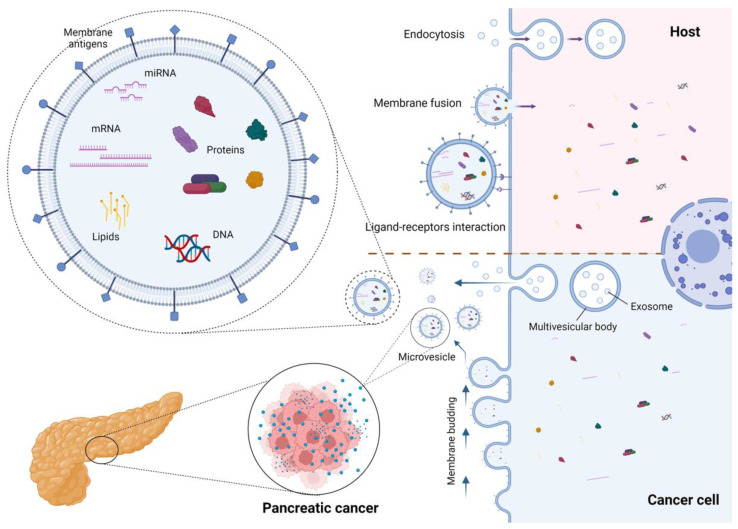
Biogenesis, structure, and cell interaction of extracellular vesicles. Created with BioRender.com.

**Figure 2 ijms-24-00885-f002:**
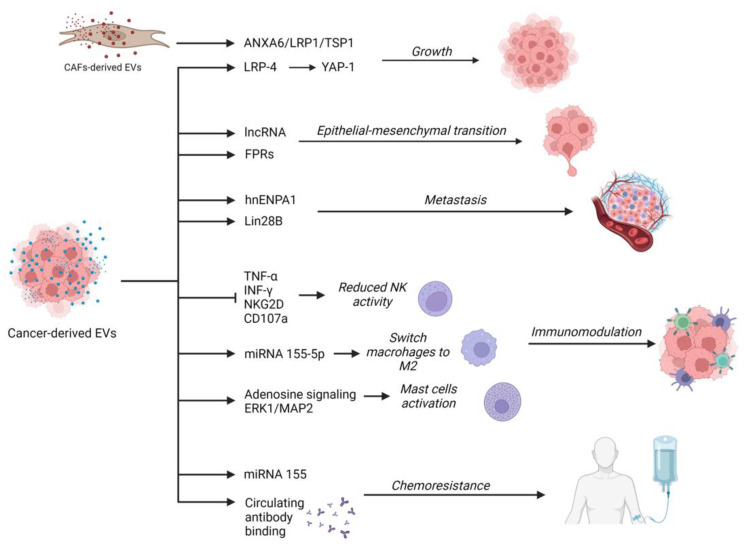
Roles of cancer-derived extracellular vesicles in pancreatic carcinogenesis, progression, and chemoresistance. Created with BioRender.com.

**Table 1 ijms-24-00885-t001:** Methods of isolation of extracellular vesicles (EVs).

Principle for EVs Isolation	Method	Type of Sample	EV Size	EV Yield	Purity	Time	Main Advantages	Main Disadvantages
Buoyant density or size-selection	Differential centrifugation [17]	Plasma Urine Cell medium	20–100 nm	Medium	Low	140–600 min	Cost	Low reproducibility Impurities Damage of exosomes Variable efficiency
	Density gradient centrifugation [18]	Urine Cell medium	20–100 nm	Medium	High	250 min–2 days	Purity	Complexity Loss of sample
	Size-exclusive chromatography [19]	Plasma Platelet-free plasma Urine Cell medium	50–500 nm	Medium	Medium	1 mL/min + column washing	Purity Reproducibility EVs integrity	Cost Complexity
	Ultrafiltration [20,21]	Plasma Urine Cell medium	50–250 nm	Low	High	130 min	Simplicity Pure preparation	Loss of sample Deformation of EVs
	Hydrostatic dialysis [22]	Urine	50–90 nm	-		30 min–1 h per 75 mL	Cost	Additional purification from bacteria
Precipitation or phase separation	Precipitation with protamine [23]	Cell culture	-	-	Low	55 min + incubation overnight	Cost Simplicity Preservation of EVs integrity	Need purification Long duration
	Precipitation with polymers [24]	Plasma (PEG) Urine, plasma, or cell medium (ExoQuick)	50–200 nm	High	Low	45–65 min	Cost (PEG) Simplicity EVs integrity	Cost with commercial kits Contamination
	Precipitation with sodium acetate [25]	Cell culture	-	-	High	130 min	Cost Simplicity EVs integrity Purity Efficiency	Contamination with non-EVs protein
	Precipitation of protein with organic solvent [26]	Plasma	20–300 nm	-	-	105 min	Cost Simplicity	Aggregation
	Two-phase isolation [27]	Plasma Mixture of exosomes and proteins	Exosomes	-	High	75–195 min	Cost Simplicity Purity Efficiency	Contamination with polymer
Affinity Interaction	Antibodies to EV receptors [28]	Cell medium Plasma	40–150 nm	Low	High	240 min	Purity High selectivity	Cost Damage of EVs
	Phosphatidylserine-binding proteins [29]	Plasma Urine Cell medium	106 nm (with Tim protein)			12 h incubation	Reversible binding Simplicity	Cost
	Heparin-modified sorbents [30]	Plasma Cell medium	-			24 h incubation	Cost EVs integrity	Need initial purification and ultracentrifugation
	Binding of heat shock proteins [31]	Plasma Urine Cell medium	30–100 nm			<1 h	EVs integrity	Cost
	Lectins [32]	Urine	-			12 h incubation	Cost Simplicity Purity	Need initial purification and ultracentrifugation
Size selection, affinity interaction	Microfluidic technologies [33]	Blood Plasma Cell culture		Medium	Low	<2 h	Rapidness Efficiency	Cost Complexity

Abbreviations: EVs = extracellular vesicles; PEG = polyethylene glycol.

## Data Availability

No new data were created in this study. All the data reported in this review were found in original articles cited in the text.

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
