# Peer review of "Diagnostic and Prognostic Role of Extracellular Vesicles in Pancreatic Cancer: Current Evidence and Future Perspectives"

_ijms, 2023, doi:10.3390/ijms24010885_

Round 1
Reviewer 1 Report
Nicoletti et al address the relevance of extracellular vesicles as a source of early diagnosis of pancreatic cancer. However, relevance and timeliness of the topic are important in the context of diagnosis and prognosis of various cancer types including pancreatic cancer.
At the same time, this manuscript needs major revision for a better impact and relevance to the readers of this journal.
Major scientific comments:
1. Introduction section needs major restructuring. Because, defining what is PC is not relevant, since literature is available. However, authors should focus on the rationale and objective of this manuscript.
2. Section 2 details about EVs and this section will be better if supported with suitable illustrations. Table on method part could be separated into a new section as Existing approaches and technologies for EVs
3. A section on Status on uses of EVs as diagnostic and prognostic markers at preclinical and clinical trials would be enriching.
4. A section on EVs and EccDNAs would be additional perspectives.
5. A separate section of EVs and metastasis would make the claim better.
6. A discussion on the export and import of EVs by cancer cells and normal cells should be presented
Minor comments: Starting from abstract, introduction till conclusion, substantial typo errors and grammatical errors are there. This makes the not so appreciable coherence of this manuscript.
Author Response
- Introduction section needs major restructuring. Because, defining what is PC is not relevant, since literature is available. However, authors should focus on the rationale and objective of this manuscript.
Re: Following the suggestion we have extrended the section on rationale and objective of the manuscript.
- Section 2 details about EVs and this section will be better if supported with suitable illustrations. Table on method part could be separated into a new section as Existing approaches and technologies for EVs
Re: We thank the reviewer for raising this point. A figure was added in order to support the explanation of EV biogenesis and cell interaction. The table on EV isolation methods was inserted to synthetize evidence on laboratory techniques, keeping the review more focused on clinical and translational aspects.
- A section on Status on uses of EVs as diagnostic and prognostic markers at preclinical and clinical trials would be enriching.
Re: We thank the reviewer for raising thi point. However in this review, we focused on clinical evidence and studies that combined preclinical bases with clinical application.
- A section on EVs and EccDNAs would be additional perspectives.
Re: Following the suggestion of the reviewer we have added a section on eccDNA in the manuscript.
- A separate section of EVs and metastasis would make the claim better.
Re: We thank the reviewer for raising this point. Evidence on EV and metastasis is discussed in sections 2, 3 and 4.
- A discussion on the export and import of EVs by cancer cells and normal cells should be presented
Re: Following the suggestion of the reviewer the export and import of EVs was discussed in the manuscript and it is also described in an additional figure.

Reviewer 2 Report
The present review manuscript titled “Diagnostic and prognostic role of extracellular vesicles in pancreatic cancer: current evidence and future perspectives” by Nicoletti et al. is novel and very well written. In this manuscript, the authors discussed almost all the aspects as per their aim and objective of the paper. However, I have some queries that are as follows.
Comment 1. As the authors stated that the production of extracellular vesicles enhanced by inflammation and tumorigenesis. This gives the basis for the diagnosis of all types of cancers. The authors should specifically stick to pancreatic cancer.
Comment 2. There is no diagrammatic representation in the manuscript. In my opinion, the authors should diagrammatically represent the role of extracellular vesicles in the progression of pancreatic cancer.
Comment 3. The authors should also discuss the patents filed for extracellular vesicles as a diagnostic marker in pancreatic cancer.
Author Response
- There is no diagrammatic representation in the manuscript. In my opinion, the authors should diagrammatically represent the role of extracellular vesicles in the progression of pancreatic cancer.
Re: According to reviewer suggestion, we added an illustration to summarize the evidence on pancreatic carcinogenesis, progression and chemoresistance.
- As the authors stated that the production of extracellular vesicles enhanced by inflammation and tumorigenesis. This gives the basis for the diagnosis of all types of cancers. The authors should specifically stick to pancreatic cancer.
Re: After a general introduction on EV classification and biogenesis, section 2 specifically focus on evidence in pancreatic cancer. The section was renamed “Extracellular vesicles in Pancreatic Cancer: Role in Carcinogenesis, Cancer Progression and Chemoresistance”.
- The authors should also discuss the patents filed for extracellular vesicles as a diagnostic marker in pancreatic cancer.
Re: According to reviewer observation we better specified this aspect in the manuscript together with the limitation of this approach for the diagnostic application of EV.

Reviewer 3 Report
The manuscript reviews the current evidence on the role of extracellular vesicles in pancreatic cancer. The authors introduce the role of extracellular vesicles in carcinogenesis, cancer progression and chemoresistance. Proteomics of extracellular vesicles in pancreatic cancer was also analyzed and reviewed, which increases the ability to screen potential candidate biomarkers. The authors list numerous studies on EV-derived miRNAs as potential biomarkers for pancreatic cancer detection. This manuscript very clearly and comprehensively describes the role and application prospects of extracellular vesicles in pancreatic cancer detection.
1. The author describes the current research status and advantages of extracellular vesicles in pancreatic cancer detection, and suggests that the author can add some clinical limitations of extracellular vesicles in detection.
2. The author cited many examples to illustrate the current research status of extracellular vesicles in pancreatic cancer. It would be more intuitive if some research-related diagrams could be added.
Author Response
- The author describes the current research status and advantages of extracellular vesicles in pancreatic cancer detection, and suggests that the author can add some clinical limitations of extracellular vesicles in detection.
Re. We change according to reviewer suggestion.
- The author cited many examples to illustrate the current research status of extracellular vesicles in pancreatic cancer. It would be more intuitive if some research-related diagrams could be added.
Re: Following the suggestion of the reviewer we added two figures to support the concepts.

Round 2
Reviewer 1 Report
The authors have improved substantially as per the suggestions.
Reviewer 2 Report
The authors addressed my comments very carefully. I don't have further comments.